# Antibody Levels Poorly Reflect on the Frequency of Memory B Cells Generated following SARS-CoV-2, Seasonal Influenza, or EBV Infection

**DOI:** 10.3390/cells11223662

**Published:** 2022-11-18

**Authors:** Carla Wolf, Sebastian Köppert, Noémi Becza, Stefanie Kuerten, Greg A. Kirchenbaum, Paul V. Lehmann

**Affiliations:** 1Research and Development, Cellular Technology Ltd. (CTL), Shaker Heights, OH 44122, USA; 2Institute of Anatomy and Cell Biology, Friedrich-Alexander University Erlangen-Nürnberg, 91054 Erlangen, Germany; 3Institute of Neuroanatomy, Medical Faculty, University of Bonn, 53115 Bonn, Germany

**Keywords:** immune monitoring, antibody-mediated immunity, ELISPOT, FluoroSpot, memory B cells, plasma cells, affinity maturation, SARS-CoV-2, influenza, EBV

## Abstract

The scope of immune monitoring is to define the existence, magnitude, and quality of immune mechanisms operational in a host. In clinical trials and praxis, the assessment of humoral immunity is commonly confined to measurements of serum antibody reactivity without accounting for the memory B cell potential. Relying on fundamentally different mechanisms, however, passive immunity conveyed by pre-existing antibodies needs to be distinguished from active B cell memory. Here, we tested whether, in healthy human individuals, the antibody titers to SARS-CoV-2, seasonal influenza, or Epstein–Barr virus antigens correlated with the frequency of recirculating memory B cells reactive with the respective antigens. Weak correlations were found. The data suggest that the assessment of humoral immunity by measurement of antibody levels does not reflect on memory B cell frequencies and thus an individual’s potential to engage in an anamnestic antibody response against the same or an antigenically related virus. Direct monitoring of the antigen-reactive memory B cell compartment is both required and feasible towards that goal.

## 1. Introduction

Antibodies and memory B cells assume fundamentally different roles in mediating humoral immunity [1]. Antibodies constitute the “first wall” of adaptive host defense and can prevent reinfection from the same (homologous/homotypic) virus. Memory B cells constitute the “second wall” and not only permit to mount an anamnestic antibody response against the homotypic virus (e.g., SARS-CoV-2 Wuhan-Hu-1 strain) after infection or vaccination, but also enable the development of a faster and more efficient antibody response to emerging heterologous viral variants that can evade the “first wall” of pre-formed antibody-mediated defenses (e.g., the Omicron variants of SARS-CoV-2). Therefore, the assessment of memory B cells can provide insights into humoral immunity that measurements of serum antibodies do not convey.

The first encounter with a virus triggers a primary immune response with naïve antigen-reactive lymphocytes clonally expanding and their daughter cells differentiating into effector and memory cells. Effector cells contribute to instant host defense, while memory cells convey the potential to engage in an accelerated and more efficient secondary immune response in case the homologous virus or a heterologous variant strain is re-encountered later in life. For T and B lymphocytes alike, effector and memory cells have unique transcriptional profiles and life spans and contribute fundamentally different functions towards maintaining host defense.

In the course of a primary B cell response, naïve B cells proliferate, undergo immunoglobulin class switching and affinity maturation while they differentiate into antibody-secreting cells (ASCs, the effectors of the B cell system), or become B memory (B_mem_) cells. Plasma cells (PC) are the prevalent ASCs in vivo. Plasma cells can be short- or long-lived, and their survival depends on competition for niches in the bone marrow (BM) [2]. The antibody molecules they secrete have a relatively short half-life of around 3 weeks [3,4]. Therefore, when antibodies are being monitored in serum (or in other bodily fluids), one needs to consider that the molecules detected are not long-lived remnants of immune memory but instead are a reflection of ongoing secretory activity in the PC compartment. After some infections, exemplified by influenza, antibody production continues in spite of the elimination of the virus and high titers of virus-reactive antibodies can be detected for decades [5,6]. After other infections, exemplified by SARS-CoV-2, virus-reactive antibody titers begin to decline within months [7,8,9]. The reason(s) why such differences in ASC activity occur are not well understood.

Antibodies are the secreted form of the B cell receptor (BCR) and convey a variety of effector functions [10,11]. Most notably, antibody binding can directly neutralize an invading virus through binding to epitopes that prevent docking and/or entry into permissive host cells. Antibody binding, along with formation of antibody–antigen immune complexes, can also trigger activation of the complement cascade and the release of proinflammatory cytokines, direct lysis of viral particles, and enhanced clearance of opsonized antigens by professional phagocytes. Furthermore, antibodies can selectively label virally infected cells and target them for destruction through antibody-dependent cellular cytotoxicity (ADCC).

In addition to the differentiation of ASCs, long-lived B_mem_ cells are also generated during the course of a primary B cell response [1,12]. Similar to their naïve precursor cells, B_mem_ cells are also resting lymphocytes that recirculate via the blood throughout the lymphoid tissues of the body and in transit can be sampled in the blood. Importantly, upon antigen re-encounter, B_mem_ cells undergo rapid and robust proliferation and give rise to new generations of daughter cell progeny that include effector cells (ASCs) and more B_mem_ cells. However, unlike their naïve precursor cells, B_mem_ cells are present in increased numbers in the body due to their clonally expanded status and have already undergone immunoglobulin class switching and affinity maturation [12,13]. Owing to this, the anamnestic response resulting from the activation of pre-existing B_mem_ cells following re-encounter with a homologous virus is not only faster and more robust, but antibodies possessing increased affinity are also produced.

Even if variants of the virus are encountered against which the pre-existing antibodies are ineffective (as is the case with the Omicron variants of SARS-CoV-2 that acquired immune evasion mutations in the spike protein, circumventing the neutralizing capacity of pre-formed antibodies elicited through natural exposure and/or vaccination with the prototype (Wuhan-Hu-1) strain [14,15]), the B_mem_ cells formed during the primary response are still likely to confer a critical immune advantage, owing to their semi-affinity-matured and clonally expanded status. Specifically, even if these B_mem_ cells express a relatively low-affinity BCR towards the variant virus initially upon re-exposure, these cells can re-enter germinal center (GC) reactions and acquire additional somatic mutations to refine and enhance their affinity towards the variant virus [16]: as the acquisition of somatic mutations is random, a subset of the B_mem_ cell repertoire elicited by the original SARS-CoV-2 infection, or through vaccination, is prone to possess an increased affinity for future antigenic variants. In this way, the immune response elicited following reinfection with an antigenically related variant virus would not start from an unselected repertoire of naïve B cells, but from clonally expanded and partially affinity-selected B_mem_ cells. Where pre-formed antibodies fail to convey protection, B_mem_ cells could still stand a chance providing a second wall of antibody-mediated defense [1].

The recent experience with the SARS-CoV-2 vaccination further highlights the importance of distinguishing between immune protection conveyed by pre-formed antibodies versus protection conferred through the recall of B_mem_ cells. Thus, while vaccination with the spike antigen induced a B and T cell response, it provided only short-term or nonreliable protection against becoming infected by SARS-CoV-2 variants: the pre-formed antibodies and effector cells could not completely prevent the virus from docking to and entering into the host’s cells to initiate viral replication. However, the severity of COVID (coronavirus disease) is largely attenuated in vaccinated individuals conceivably due to their memory cells’ ability to rapidly re-engage into secondary heterosubtype-specific immune responses endowing a critical host defense advantage. Immune monitoring that is confined to measurements of serum/plasma antibody reactivity alone would therefore not suffice for predicting the vaccine’s efficacy without accounting for the memory cell compartment.

The presence of virus-reactive antibodies and their neutralizing potential can readily be measured using a number of well-established techniques [17,18,19]. In contrast, the detection and study of virus-reactive B_mem_ cells have been a major challenge, primarily because B cells reactive for any given antigen exist at very rare frequencies among all B cells, which themselves constitute only a fraction of all peripheral blood mononuclear cells (PBMC). Consequently, there is an abundance of publications on serum antibody reactivity and the neutralizing capacity against SARS-CoV-2 [7,9,18,20,21,22,23], but the number of successful B_mem_ studies is more limited for this virus [24,25,26,27,28,29,30] and other viruses as well [31,32,33,34,35]. Efforts to elucidate the antigen-reactive B_mem_ cell compartment have relied on the use of fluorescently labeled antigens in conjunction with flow cytometry, the assessment of antibody reactivity in culture supernatants from limiting dilution cultures of polyclonally stimulated PBMCs, or on the ELISPOT/FluoroSpot technology [36]. As recirculating B_mem_ in the blood does not secrete antibodies spontaneously, they need to be preactivated to promote differentiation into ASCs before they can be detected in either a limiting dilution or ELISPOT/FluoroSpot assays. Based on the following compelling arguments in favor of ELISPOT/FluoroSpot, we selected to refine this technique so it can become suitable for monitoring B_mem_ cells that are reactive with SARS-CoV-2 or other viral antigens. The ELISPOT/FluoroSpot approach’s main strength is that it enables the enumeration of antigen-reactive memory B cells in the blood, providing insights into the magnitude of the memory B cell pool and also of the antibody classes/subclasses produced by each memory cell, thus revealing the quality of B cell memory. Another advantage of ImmunoSpot assays is that they can be conducted with unseparated PBMCs, without the need for prior B cell isolation/sorting as B cells are the sole source of secreted antibodies in PBMCs (the need for B cell enrichment/sorting would significantly constrain assay throughput for immune monitoring purposes). A further strength of the ELISPOT/FluoroSpot test system is that, once the protocols have been established for a given antigen, it can be readily implemented for high-throughput regulated testing, as required for clinical trials [37]. One of the weaknesses of this test system is that it requires the use of perishable live blood cells. However, this obstacle can be overcome through well-established cryopreservation and thawing strategies [38,39]. A further weakness of this test system is that the phenotypic markers of the antigen-specific memory B cells are not revealed [36].

In traditional B cell ELISPOT/FluoroSpot assays, the membrane is coated with an antigen of interest, e.g., a viral protein. Onto this antigen-coated membrane, the test subject’s PBMCs are seeded, containing the rare antigen-reactive ASCs of interest. In the absence of recent antigen encounters, however, B_mem_ cells exist in a resting state and do not constitutively secrete antibodies. To overcome this obstacle, PBMCs need first to be activated through polyclonal stimulation to drive the terminal differentiation of B_mem_ cells into ASCs prior to their measurement in the assay [40,41,42]. While most B cells can be transitioned into ASCs following polyclonal stimulation protocols, each secreting large quantities of its encoded BCR as soluble antibodies, only the ASCs producing antibodies with a sufficient binding affinity for the coated antigen will leave a detectable antibody footprint on the antigen-coated assay membrane. As such, the number of spot-forming units (SFU) in a well reveals the number of antigen-reactive ASCs present in the well versus all ASCs present in that well. In this way, the magnitude of the antigen-reactive B_mem_ cell compartment can be established. The immunoglobulin class and subclass produced by each ASC can also be defined through the use of class/subclass-specific detection reagents [43]. The visualization of the plate-bound analytes relies either on the use of precipitating substrates (ELISPOT) or on fluorescence (FluoroSpot). Other than the detection step, ELISPOT and FluoroSpot are essentially the same; jointly, we refer to them as ImmunoSpot^®^.

The classical B cell ELISPOT assay was introduced decades ago [44,45], but since then, it has only been implemented successfully for a few human immune monitoring efforts [46] due to the simple reason that, for most antigens, such assays could not be established. This is because absorption of the antigen to the membrane is governed by weak, nonspecific binding forces such as hydrophobicity and charge. Successful coating therefore depends on the physical properties of the membrane chosen versus the corresponding properties of the antigen itself. To overcome this limitation in the development of B cell ImmunoSpot^®^ assays for “any antigen of interest”, we recently introduced an approach termed “affinity capture coating” (ACC) in which improved antigen absorption to the membrane is achieved through preconditioning the assay membrane with an antibody specific for the genetically encoded hexahistidine (6XHis) affinity tag of the recombinant proteins [47], enabling the subsequent high-affinity capture of any 6XHis-tagged protein. Using this universal ACC approach, we developed ImmunoSpot^®^ assays that enable the detection of memory B cells reactive with antigens representing SARS-CoV-2, seasonal influenza, and Epstein–Barr virus (EBV) [47].

Access to the above antigen-specific B cell ImmunoSpot^®^ assays permitted us in the present study to ask a fundamental question that (due to the limitations in detecting antigen-reactive B_mem_ cells so far) has not been systematically addressed: how do antibody levels reflect on B_mem_ cell frequencies reactive with a given antigen? In other words, do high antibody levels in an individual indicate that this person also has high numbers of B_mem_ cells, as intuition would suggest? Therefore, if an individual is more protected from becoming reinfected owing to his/her higher titer of pre-existing antibodies than another individual who developed a lower antibody titer, is the former also prone to mount a stronger anamnestic antibody response following re-infection? Do low/negative antibody titers in an exposed/vaccinated individual imply that this individual did not possibly develop a stronger B cell memory response than an individual with high antibody titers?

We studied here the relationship between antibody levels and B_mem_ cell frequencies for viral infections that fundamentally differ in terms of verifiable exposure. One cohort consisted of unvaccinated subjects that were infected with the SARS-CoV-2 virus, and these samples were collected after complete convalescence. A second cohort consisted of subjects whose plasma and PBMCs were cryopreserved in the pre-COVID era and served as an unambiguous negative control group for SARS-CoV-2 immunity. The importance of the latter control group is highlighted by the fact that for the other viruses we studied (influenza or EBV), unambiguous negative controls cannot be established as most adults in the human population have been exposed to them [5,48]. Calling into question whether antibody reactivity is a reliable indicator of antigen exposure and immunity, in a previous study [49], we established that the majority of human cytomegalovirus (HCMV) seronegative donors indeed have high numbers of HCMV antigen-reactive CD4^+^ and CD8^+^ memory T cells as well as B_mem_ cells. The present study will confirm and extend this notion.

The three viral infections we studied here for the relationship between antibody levels and B_mem_ cell frequencies also fundamentally differ in terms of duration and the periodicity of antigen exposure. The SARS-CoV-2 infection results in clearance of the virus by the time of recovery from the infection, typically within two weeks. Therefore, the subjects in our SARS-CoV-2 cohort had a single, full-blown exposure to this virus without any subsequent antigen stimulation. EBV, in contrast, is not cleared following resolution of the acute infection and instead persists latently in B cells causing occasional subclinical viral flares and repetitive and potentially chronic antigen stimulation [50,51]. Influenza, similar to SARS-CoV-2, is also rapidly cleared following initiation of a successful immune response. However, the influenza exposure histories of our donor cohorts, and specifically their circulating antibodies and antigen-reactive B and T cell repertoires, are far more complex, owing to the contributions of prior infections and/or vaccinations over their respective lifetimes [52,53].

Studying these three fundamentally different viruses and immune scenarios for the respective antigen-reactive antibody levels by ELISA and the B_mem_ frequency in PBMCs by ImmunoSpot^®^, we show that, in each of these antigen systems, these two immune parameters do not strongly correlate. Discordance between pre-formed humoral immunity and B cell memory potential, therefore, might be common in antiviral immunity.

## 2. Materials and Methods

### 2.1. Human Subjects

PBMC and plasma samples from pre-COVID-19-era donors were obtained from healthy adults and originated from CTL’s ePBMC^®^ library (CTL, Shaker Heights, OH, USA). Samples (*n* = 54) were collected at FDA-registered collection centers from IRB-consented healthy human donors by leukapheresis and then were sold to CTL identifying donors by code only, while concealing the subjects’ identities. PBMCs were cryopreserved and stored in liquid nitrogen until testing. Plasma was aliquoted and stored at −20 °C until testing. Additional information pertaining to the pre-COVID-19 donors is provided in Appendix A. Importantly, these pre-COVID-19 samples serve as an unambiguous control group for SARS-CoV-2 immunity since at the time of their collection (prior to 1 November 2019), the SARS-CoV-2 virus had not yet begun circulating in the USA, and COVID-19 vaccines were not available. 

The SARS-CoV-2 infection-verified blood samples were collected between April and October 2020. However, based on PCR testing, these infections occurred in the months of April, May, and June when the initial Wuhan-strain virus was circulating and prior to the emergence of new SARS-CoV-2 variants or the availability of COVID-19 vaccines. Accordingly, we evaluated B cell and circulating antibody reactivity against the spike protein encoded by the Wuhan-strain virus. This simple, defined, immunologic scenario was chosen to avoid the introduction of further immune variables caused by repeat infections with partially cross-reactive heterotypic virus variants in a mostly (homotypic spike antigen) vaccinated population in which reinfections frequently are subclinical or undiagnosed.

PBMC and plasma samples comprising the convalescent COVID-19 donor cohort (*n* = 25) were obtained from three sources as follows: five donors were purchased from the Oklahoma Blood Institute (Oklahoma city, OK, USA) and were received as previously cryopreserved PBMCs and plasma aliquots; ten donors were recruited by the American Red Cross (Atlanta, GA, USA), BioIVT (Westbury, NY, USA), or Stem Express (Folsom, CA, USA) with IRB approval and then were sold to CTL identifying donors by code only while concealing the subjects’ identities; and ten donors were collected internally at CTL under an Advarra Approved IRB #Pro00043178 (CTL study number: GL20-16 entitled COVID-19 Immune Response Evaluation). All PBMCs were stored in liquid nitrogen until testing. Plasma was aliquoted and stored at −20 °C until testing. All subjects included in the COVID-19 cohort tested positive for SARS-CoV-2 RNA by PCR. Additional information on the COVID-19 donors is provided in Appendix A.

### 2.2. Polyclonal B Cell Stimulation

Detailed methods of thawing, washing, and counting PBMCs have been previously described [39]. The freshly thawed PBMC samples were resuspended in B cell medium (BCM) containing RPMI 1640 (Lonza, Walkersville, MD, USA) supplemented with 10% fetal bovine serum (Gemini Bioproducts, West Sacramento, CA, USA), 100 U/mL penicillin, 100 U/mL streptomycin, 2 mM L-Glutamine, 1mM sodium pyruvate, 8 mM HEPES (all from Life Technologies, Grand Island, NY, USA), and 50 µM β-mercaptoethanol (Sigma-Aldrich, St. Louis, MO, USA). PBMCs were then stimulated with human B-Poly-S (CTL) at 2 × 10^6^ cells/mL in 25 cm^2^ tissue culture flasks (Corning, Sigma-Aldrich) and incubated at 37 °C, 5% CO_2_ for five days to drive the differentiation of resting memory B cells into ASCs prior to evaluation in ImmunoSpot^®^ assays.

The five-day polyclonal B cell activation culture needed for the detection of B_mem_ cells via their secretory antibody footprint involves B cell proliferation, and thus it can be expected that the absolute numbers of B_mem_ cells recorded after such a cell culture will be slightly increased relative to the numbers of B_mem_ cells present in the blood ex vivo. 

### 2.3. Recombinant Proteins

Our objective was to monitor antibody and B cell responses that would be specific to SARS-CoV-2. Since the S2 subunit (S2) of the spike protein is more conserved between seasonal coronaviruses and SARS-CoV-2, we focused on the S1 domain. SARS-CoV-2 spike subunit 1 (S1) was purchased from Creative Diagnostics (Shirley, NY, USA). The SARS-CoV-2 nucleocapsid (NCAP) was purchased from RayBiotech (Peachtree Corners, GA, USA). The recombinant Epstein–Barr virus nuclear antigen 1 (EBNA1) protein was purchased from Serion (Würzburg, Germany). Recombinant hemagglutinin (rHA) proteins encoding A/California/04/2009 (CA/09, H1N1), A/Texas/50/2012 (TX/12, H3N2), and B/Phuket/3073/2013 (Phuket/13, FluB) seasonal influenza vaccine strains were acquired from the Center for Vaccines and Immunology (CVI) (University of Georgia (UGA), Athens, GA, USA) and have been described previously [54,55]. Importantly, all recombinant proteins used in this study possessed a genetically encoded 6XHis affinity tag.

### 2.4. B Cell ImmunoSpot^®^ Assays

Following polyclonal stimulation, PBMCs were harvested from tissue culture flasks and washed with PBS prior to counting using CTL’s Live/Dead cell-counting suite on an ImmunoSpot^®^ Ultimate S6 Analyzer. Cell pellets were resuspended at 1–3 × 10^6^ live cells per mL (when measuring antigen-reactive IgG^+^ ASCs) or 3 × 10^5^ live cells per mL (for the detection of all IgG-secreting cells) in complete BCM and seeded into ImmunoSpot^®^ assays.

Unlike memory B cells that express an IgM BCR, the memory B cell population that primarily participates in recall responses has already undergone class-switch recombination and is programed to secrete IgG [1]. In this study, we therefore focused on the IgG-secreting memory B cell population. For the enumeration of all IgG-secreting cells (total IgG^+^ ASCs), the cell suspensions were serially diluted 2-fold in duplicates starting at 3 × 10^4^ live cells/well in round-bottom 96-well tissue culture plates (Corning, Sigma-Aldrich). Subsequently, cells were transferred into ImmunoSpot^®^ assay plates that were precoated with anti-κ/λ capture antibody reagents (from CTL) and incubated for 16 h at 37 °C, 5% CO_2_. Plate-bound immunoglobulin (Ig) spot-forming units (SFU) were subsequently visualized using a human IgG-detecting ImmunoSpot^®^ kit (from CTL) according to the manufacturer’s instructions.

For the enumeration of antigen-reactive IgG^+^ ASCs, ImmunoSpot^®^ assays were performed with ACC as previously described [47] and illustrated by Appendix A. Briefly, assay plates were first preconditioned with 70% (*v*/*v*) EtOH followed by two washing steps with PBS. Next, wells were coated with purified anti-6XHis-tag antibodies (BioLegend, San Diego, CA, USA) at 10 μg/mL in PBS overnight at 4 °C. The following day, assay plates were washed once with PBS and then coated overnight with 6XHis-tag-labeled recombinant proteins at 10 μg/mL in PBS. After overnight incubation of the 6XHis-tagged recombinant protein coating solutions at 4 °C, plates were washed once with PBS and then blocked with complete BCM for 1 h at room temperature prior to the addition of polyclonally stimulated PBMCs at the specified cell numbers per well. Plates were incubated at 37 °C, 5% CO_2_ for 16 h, and SFUs were subsequently visualized using the human IgG-detecting ImmunoSpot^®^ kit (from CTL) according to the manufacturer’s instructions. This antigen coating protocol avoids the direct capture of ASC-derived antibodies on the PVDF membrane while assuring the selective coating reagent-mediated capture of ASC-derived antibodies (Appendix A).

### 2.5. ImmunoSpot^®^ Image Acquisition and SFU Counting

Plates were air-dried prior to scanning on an ImmunoSpot^®^ Ultimate S6 Analyzer. Total or antigen-reactive IgG^+^ SFUs were then enumerated using the Fluoro-X suite and the basic count mode of the ImmunoSpot^®^ Software (Version 7.0.27). To account for the variable starting frequencies and the differentiation of memory B cells during the polyclonal stimulation protocol, data are reported as the frequency of antigen-reactive ASCs among total IgG^+^ ASCs [56]. ImmunoSpot^®^ B cell kits, analyzers, and software proprietary to CTL were used in this study; we refer to this collective methodology as ImmunoSpot^®^.

For the accurate counting of SFUs per well, we relied on a serial dilution strategy shown in Figure 1A, Appendix A and detailed in Section 3. As shown, at low SFU numbers per well, there was a linear relationship between the cell numbers plated and SFUs counted from which, via linear regression and extrapolation, the frequencies of the ASCs in PBMCs could be established. For routine testing, our ImmunoSpot^®^ assay approach was tailored to achieve maximal sensitivity in detecting low-frequency memory B cell responses, and thus a single (1–3 × 10^5^) cell per well input was used. In such cases, SFU counts in the critical <100 SFU/well range were accurate; however, higher SFU counts per well approached the “upper bound” of accurate enumeration and therefore likely underrepresent the actual antigen-reactive B cell frequency. Following conventions in flow cytometry, the data are expressed as the percentage of antigen-reactive SFUs among all IgG^+^ ASCs.

### 2.6. Bivariate Visualization of FluoroSpots

The counted FluoroSpots from replicate wells of an individual donor originating from the same cell input were merged into flow cytometry standard (FCS) files using the ImmunoSpot^®^ Software (Version 7.0.27.0) and were visualized using Flowjo^TM^ (Version 10.6.2) (Ashland, OR, USA).

### 2.7. ELISA Assays

To evaluate IgG reactivity in plasma samples, Immulon^®^ 4HBX plates (Thermo Fisher Scientific, Waltham, MA, USA) were coated overnight at 4 °C with 2 μg/mL of the respective recombinant antigens: rHA representing CA/09, TX/12 or Phuket/13 of seasonal influenza strains, the EBNA1 protein in carbonate buffer (pH 9.4), the SARS-CoV-2 S1, or the SARS-CoV-2 NCAP proteins in PBS. After blocking the plates with the ELISA blocking buffer containing 2% *v*/*v* BSA in PBS with 0.1% *v*/*v* Tween20 (Sigma-Aldrich) for 1 h at room temperature, plasma samples were serially diluted and incubated for 2 h. Following four washes with 150 μL of PBS, horseradish peroxidase-conjugated anti-human IgG detection reagent (from CTL) was added to the plates and incubated for 1 h at room temperature. After washing the plates four times with PBS, 100 μL of TMB chromogen solution (Thermo Fisher Scientific, Waltham, MA, USA) was added to develop the assay. Conversion of the TMB substrate was terminated by the addition of 100 μL/well of 1M HCl, and the optical density was measured at 450nm (OD_450_) using a Spectra Max 190 plate reader (Molecular Devices, San Jose, CA, USA). The abundance of IgG reactivity against the respective antigens was then interpolated into μg/mL of IgG equivalents using SpotStat^TM^ (Version 1.6.4.0, CTL) based on standard curves generated by directly coating decreasing quantities of a reference IgG preparation (from Athens Research and Technology, Athens, GA, USA) in duplicates into designated wells of each assay plate.

### 2.8. Statistical Methods

Student’s *t*-tests were used to evaluate differences in the serum antibody reactivity or antigen-reactive B cell frequencies between the pre-COVID-19 and COVID-19 donor cohorts (GraphPad Prism Version 9.2, San Diego, CA, USA). Pearson correlation analysis was performed using the GraphPad Prism software on log-transformed antigen-reactive ASC data and the corresponding antigen-reactive IgG titers (expressed as μg/mL of IgG equivalents). Regressions were also performed using GraphPad Prism and the *R*^2^ values, 95% confidence bands, and *p*-values less than 0.05 were plotted in the corresponding figures.

## 3. Results and Discussion

The primary goal of this study was to test the hypothesis that virus-reactive antibody levels reflect on the frequency of B_mem_ cells. Based on long held “textbook knowledge”, such a close correlation could be expected as (a) plasma cells are thought to be long-lived and to continuously produce antibodies, (b) B_mem_ cells are also assumed to be long-lived, and (c) both cell types were thought to arise during the course of a B cell immune response at a constant ratio from a common precursor cell. The recent literature discussed below, including the data presented in the following communication, call into question that such a direct correlation would be the rule after immune responses to viral infections.

Our first concern addressing this hypothesis was to optimize the accuracy of measuring both antigen-reactive antibody levels and the frequency of antigen-reactive B_mem_ cells. For the former, we performed standard ELISAs involving serial dilutions of the donor plasma samples in conjunction with a “μg/mL-equivalent scale” for analyzing the data [57,58,59,60]. This approach is superior to the “area under the curve” approach as it leverages an internal (plate-specific) reference standard and thus is independent of assay-associated variability related to development time and temperature.

While the methodology for the quantification of antigen-reactive antibodies in plasma by ELISA is well-established, we first needed to develop such for the objective and accurate counting of antigen-reactive B_mem_ cells in the blood. Empirically, antigen-reactive B cell ImmunoSpot^®^ assays result in diverse spot morphologies [47] (see also Figure 1B,C). This outcome is expected as the secretory footprints of individual ASCs are defined by a multitude of parameters [61] including (a) the net amount of antibodies produced by the individual B cells during the assay’s duration, (b) the kinetics of antibody production by the ASCs, and (c) the functional affinity of each ASC’s antibody for the antigen (that can encompass a broad spectrum among an antigen-reactive B cell repertoire, defining the capture and dissociation rates of antibodies binding to antigens). Furthermore, pan-well or regional “ELISA” effects modulate the background membrane staining when an antibody that “escaped” into the supernatant is captured distally from the source ASC. The crowding of spots also interferes with unambiguous counting. For all these reasons, the first part of this manuscript is dedicated to establishing unambiguous frequencies of antigen-reactive B cells in PBMCs.

### 3.1. Establishing Unambiguous Frequencies of Antigen-Reactive B Cells in PBMCs

The accurate detection and counting of antigen-reactive B cells in PBMCs predicts a linear relationship between the cell numbers plated in the ImmunoSpot^®^ assay and the antigen-reactive SFUs counted. This is because B cells are the only cell type that secretes antibodies and because each B cell produces antibodies with pre-defined specificity. For each antigen reported in this study, we have established the ideal test conditions for accurate frequency measurements. PBMCs were polyclonally activated with R848 and IL-2 for five days to promote the differentiation of resting B_mem_ cells (which do not secrete antibodies) into ASCs that can be detected in ImmunoSpot^®^ assays via their secretory footprints [40,41,42]. Such pre-stimulated PBMCs were plated in serial dilutions into antigen-coated assay plates while increasing the number of replicate wells at lower cell numbers to account for the expected Poisson variation. Appendix A shows the raw data for such experiments; in the example shown, two representative donors that have recovered from COVID-19 were evaluated for ASC reactivity against the SARS-CoV-2 S1 protein.

As expected, the number of detectable SFUs decreased with the number of stimulated PBMCs plated (Figure 1A and Appendix A). However, the secretory footprints of individual ASCs were not readily resolved at the highest cell inputs due to the confluence of spots and elevated background membrane staining arising as a consequence of the elevated antigen-reactive ASC frequency. As expected, at lower cell inputs, the footprints of individual ASCs became clearly discernable and facilitated the accurate enumeration of individual antigen-reactive ASCs in such wells (Figure 1B). In addition, as expected, the SFUs displayed a wide spectrum of sizes and fluorescent intensities (Figure 1B,C), reflecting on the individual ASC’s range of productivity and functional affinity for the SARS-CoV-2 S1 protein. Therefore, when we used machine reading for the enumeration of SFUs, we counted all SFUs, including all sizes and densities. The counting of such “ungated events” showed a close to perfect linear relationship between SFU counts and cell numbers plated per well for (but only for) wells that contained low numbers of SFUs. For COVID-19 Donor 1, for example, these were wells containing ≤2 × 10^4^ PBMC/well (Figure 1E). For this donor, SFU counts for cell numbers exceeding 2 × 10^4^ PBMC/well were reduced below the expected count (Figure 1D). Once the frequency-dependent linear range was established for a subject, linear regression permitted reliable calculations of the frequency of antigen-reactive B cells within PBMCs. For COVID-19 Donor 1 this number was computed to be 195 SARS-CoV-2 S1-reactive ASCs/10^5^ PBMCs, and for COVID-19 Donor 2, providing another example, it was 163 S1-reactive ASCs/10^5^ PBMCs (Appendix A).

The above data establish that the accurate counting of antigen-reactive B cells in ImmunoSpot^®^ assays requires the establishment of the range in which SFU counts and cell inputs per well exhibit a linear relationship from which frequencies can be reliably extrapolated. This is critical as the number of antigen-reactive B cells can span several orders of magnitudes, even between antigen-primed individuals (see below). We optimized test conditions and verified these basic assumptions for the accurate counting of SFUs for all antigen-reactive B cell ImmunoSpot^®^ assays reported in this publication; representative results from such experiments are shown in Appendix A.

As a positive control, we also performed total IgG^+^ ImmunoSpot^®^ assays in parallel to verify the functionality of the B cells following their in vitro stimulation. In this assay variant, the membrane is coated with anti-human immunoglobulin light chain (IgL) (anti-Igκ/Igλ) capture antibodies instead of the antigen itself, capturing the secretory footprint of all ASCs irrespective of their binding reactivity. Applying the rules described above, we have established the frequency of the total IgG^+^ ASCs in each test subject through serial dilution (the raw data and results for six representative samples are depicted in Appendix A). Because the frequency of the total IgG^+^ ASCs exhibited considerable inter-individual variations among the test subjects following in vitro stimulation, when frequencies of antigen-reactive IgG^+^ ASCs are reported in the following, they are expressed as a percentage of the total IgG^+^ ASC compartment, i.e., “% antigen-reactive IgG^+^ ASC” [56].

### 3.2. Equal Overall Assay Performance of Pre- and Post-COVID-19 PBMCs

SARS-CoV-2 exposure, unlike the endemic seasonal influenza and EBV infections also studied here, is unique in as much that the exposure can be verified beyond measuring antibody reactivity. As the first laboratory confirmed case of the SARS-CoV-2 virus infection in the United States occurred on 20 January 2020 [62], PBMCs collected before 1 November 2019 have been verifiably derived from subjects who could not have been exposed to SARS-CoV-2 and thus must be immunologically naïve to this virus since COVID-19 vaccines were not yet available. It is a matter of debate, however, how much T and B cell cross-reactivity exists in pre-COVID-19-era subjects elicited through prior exposure(s) to endemic coronavirus strains [63,64,65,66] and possibly other antigens [67]. It also remains unclear whether such pre-existing, cross-reactive immunity contributes to defining the severity of the primary SARS-CoV-2 infection. In addition, unlike for influenza and EBV, not only the exposure itself, but even the time point of the primary SARS-CoV-2 infection can be verified by PCR testing. For this study, we therefore could compare B cell reactivity to SARS-CoV-2 antigens in two cohorts that are highly defined with respect to immune exposure to this virus: PBMCs collected from individuals who could not have been infected (the “pre-COVID-19” cohort) and those who were collected after a PCR-verified SARS-CoV-2 infection during the first wave in 2020, caused most likely by the original “prototype” Wuhan-Hu-1 strain (the “COVID-19” cohort). Notably, all PBMCs were collected before the COVID-19 vaccination became available.

PBMCs of the test subjects were cryopreserved according to a protocol that maintains B cell functionality [38]. As shown in Appendix A, the viability of the PBMCs was comparable for both cohorts after thawing. The functionality, the number of the total IgG^+^ ASCs, while showing the expected inter-individual variations within each group, was comparable among the two cohorts (Appendix A). When tested in two separate experiments, the frequencies of the total IgG^+^ ASCs reproduced well for the individual donors (Appendix A), suggesting that the inter-donor variability seen is inherent to each donor and does not represent an assay variable. Appendix A depicts the correlation between the PBMC viability after five days of polyclonal stimulation for each donor in both cohorts plotted versus the total IgG^+^ ASC frequency (that is, “what percentage of viable cells were IgG^+^ ASCs”). While, as expected, there was no direct correlation between the overall viability of the PBMCs and total IgG^+^ ASC frequency, the cells from both cohorts behaved similarly. Collectively, these observations suggest that if any difference are seen in the antigen-reactive frequencies among these cohorts, they cannot be attributed to the freezing conditions or the duration of time the cells were stored under liquid nitrogen (the comparison of influenza- and EBV-reactive B cell frequencies between the two cohorts detailed later in this communication, and the inter-assay reproducibility of both the total and antigen-reactive IgG^+^ ASC frequencies presented in Appendix A, respectively, will further substantiate this claim).

### 3.3. Exquisite Specificity of SARS-CoV-2 Antigen-Reactive B Cell ImmunoSpot^®^ Assays

With the exception of a single donor, the PBMCs from all other subjects in the COVID-19 cohort (*n =* 25) collected following convalescence from the SARS-CoV-2 infection displayed S1-reactive IgG^+^ ASCs after polyclonal stimulation, albeit in frequencies that spanned three orders of magnitude, ranging between 0.01% and 1.15% of all IgG^+^ ASCs (Figure 2A). In contrast, less than five S1-reactive IgG^+^ ASCs were detected in any of the pre-COVID-19-era donors, even at the highest cell number tested (3 × 10^5^ PBMC/well). Importantly, at this high cell input, such S1-reactive IgG^+^ ASCs were often too numerous to accurately count for several COVID-19 donors (Appendix A).

Testing for B_mem_ cell reactivity against the SARS-CoV-2 nucleocapsid (NCAP) protein reproduced the exquisite specificity seen for the S1 antigen (Figure 3A). As with the S1 antigen, NCAP-reactive ASCs occurred in frequencies less than 5 SFUs per 3 × 10^5^ PBMCs for all pre-COVID-19 era donors tested. In stark contrast, IgG^+^ ASCs with reactivity against seasonal influenza and/or EBV antigens were readily detectable in pre-COVID-19 donors (Figure 4 and Appendix A). The absence of detectable NCAP-reactive IgG^+^ ASCs in pre-COVID-19 donors is of particular interest: first, because the NCAP proteins expressed by novel SARS-CoV-2 virus strains share a greater sequence conservation and predicted immunogenic epitopes with circulating seasonal coronaviruses than the spike (S1) protein and as such are more likely to cross-react with B cells triggered by seasonal coronaviruses [68,69], and second, because establishing immunity to NCAPs is gaining importance for the immunodiagnostics of the SARS-CoV-2 infection in COVID-19 vaccinated (i.e., spike antigen immunized) individuals.

### 3.4. Discordance between SARS-CoV-2 Antigen-Reactive Memory B Cell Frequencies and Antibody Levels

We also performed ELISA assays involving the same recombinant SARS-CoV-2 antigen preparations as applied above for ImmunoSpot^®^ testing. Plasma from both cohorts were tested in serial dilutions alongside an IgG reference standard to establish for each subject the abundance of S1 (Figure 2B) and NCAP (Figure 3B) antigen-reactive IgGs. Judged as cohorts, the results were clear-cut and revealed a significantly increased IgG reactivity against the SARS-CoV-2 S1 and NCAP coating antigens. However, the results were less clear-cut when judged at the level of individual subjects. Approximately half of the subjects in the COVID-19 cohort showed low levels of IgG reactivity against either the S1 or NCAP coating antigens while displaying elevated frequencies of antigen-reactive IgG^+^ ASCs (derived from memory B cells) following in vitro stimulation (Figure 2C and Figure 3C). Thus, for immunodiagnostic purposes, the detection of B cell memory appears to be a more sensitive and reliable indicator of SARS-CoV-2 infection history than measurements of antibody reactivities alone.

**Figure 3 cells-11-03662-f003:**
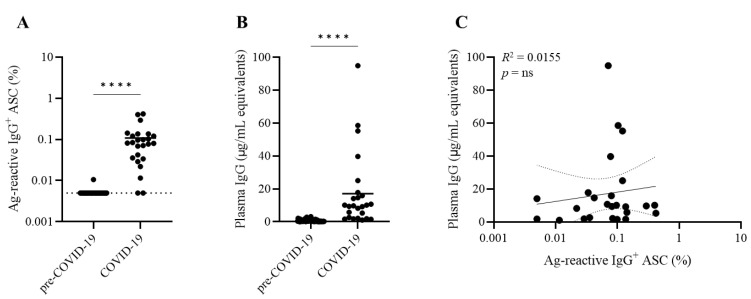
Specificity of SARS-CoV-2 nucleocapsid (NCAP)-reactive IgG^+^ memory B cell detection and discordance between circulating antibody levels and memory B cell frequencies in PBMCs. (**A**) SARS-CoV-2 NCAP-reactive IgG^+^ ASC frequencies in PBMCs from pre-COVID-19 (*n* = 54) or convalescent COVID-19 donors (*n* = 25). Donor PBMCs were pre-stimulated for 5 days in vitro and were seeded into SARS-CoV-2 NCAP-coated wells as described in Appendix A and Materials and Methods. For each donor, represented by an individual dot, the frequency of NCAP-reactive ASCs is expressed as the percentage of the total IgG^+^ ASC compartment in that subject. Dotted line denotes limit lower limit of quantification. Statistical significance was determined using an unpaired Student’s *t*-test. **** *p* < 0.0001. (**B**) SARS-CoV-2 NCAP-reactive circulating IgG antibody levels in pre-COVID-19 and COVID-19 donors were measured by ELISA (detailed in Materials and Methods) alongside an IgG reference standard that allowed for the interpolation of NCAP-reactive IgG levels as “μg/mL equivalents”. Donors are represented as individual dots. Solid line denotes mean IgG level of the COVID-19 donor cohort. Statistical significance was determined using an unpaired Student’s *t*-test. **** *p* < 0.0001. (**C**) Correlation between SARS-CoV-2 NCAP-reactive IgG^+^ ASC frequencies (x-axis) and circulating IgG levels (y-axis) in donors comprising the COVID-19 cohort.

For both the SARS-CoV-2 S1 and NCAP antigens, plasma IgG levels and frequencies of antigen-reactive IgG^+^ ASCs (derived from B_mem_ cells recirculating in blood) were poorly correlated. Specifically, the *R*^2^ values were 0.2535 for the S1 protein (Figure 2C) and 0.0155 for NCAP (Figure 3C), respectively. Therefore, in the context of SARS-CoV-2 immunity, plasma antibody levels were poor indicators of the corresponding antigen-reactive memory B cell pool sizes. These measurements were made on blood samples collected within months after recovery from COVID-19. As antibody titers are lower after mild compared to severe SARS-CoV-2 infections [70,71] and additionally wane over time [7,8,9], while B_mem_ cell frequencies are thought to be more stable over time, one might expect the discordance to grow as time passes, but longer-term longitudinal studies are needed to address this point. Irrespective of the outcome of those future studies, however, the main message of this communication is likely to hold up: the detection of B_mem_ cells themselves is likely to be a more sensitive and reliable indicator of immune exposure and memory potential than standard measurements of antibody reactivities alone, and antibody levels are poor indicators of memory B cell frequencies in any given individual.

### 3.5. Discordance between Influenza Virus and EBV Antigen-Reactive Memory B Cell Frequencies and Antibody Levels

In the next set of data, we aimed to establish whether the findings reported above for SARS-CoV-2 are unique to this viral infection and/or the circumstances of our testing. The IgG^+^ B cell response induced by the SARS-CoV-2 infection might be unique due to the immune evasion strategies of the virus [72]. However, even if that is not the case, the dissociation between plasma antibody levels and B_mem_ cell frequencies could just be a feature of (a) a primary B cell response seen (b) after recovery from a mild infection with a virus that (c) typically is cleared within 2 weeks, as these all apply to the COVID-19 cohort tested above. Studying the influenza-reactive B cell immunity might therefore provide further insights in this regard. The influenza virus is also cleared by the immune system (with rare exceptions of severe infections) within 2 weeks post-infection. However, the time point of the primary infection likely lies in the distant past, years, or even decades ago, and reinfections causing secondary B cell responses can be assumed for most adults. As seen in Figure 4A–C, the correlation between plasma antibody levels and B_mem_ cell frequencies was only marginal for three representative influenza virus strains, with *R*^2^ values of 0.1896 for CA/09 (H1N1), 0.0789 for TX/12 (H3N2), and 0.0913 for Phuket/13 (FluB) rHA antigens, respectively. These data suggest that the observations reported above for SARS-CoV-2 are neither unique to this virus, nor related to the early time point of the measurements but might be a more universal feature of the B cell response following (at a minimum) respiratory tract infections.

Infection with EBV represents yet a fundamentally different immunological scenario. The primary infection occurs mostly during young adulthood (EBV infection is also called “student kiss fever”) and is systemic. Furthermore, unlike SARS-CoV-2 and influenza, EBV is not cleared from the body but persists lifelong with occasional reactivation episodes, and EBNA1 is expressed during latency programs I, II, and III [73,74]. As seen in Figure 4D, the correlation between EBNA1-reactive IgG antibody levels in plasma and IgG^+^ memory B cell frequencies was only marginal in healthy adult donors, with an *R*^2^ value of 0.1732. Additionally important for immune diagnostic purposes, such as with SARS-CoV-2 and influenza, several subjects exhibited low levels of plasma antibody reactivity against the EBNA1 protein, a finding that when viewed in isolation could be suggestive of either a lack of virus exposure or of developing a weak B cell response to the exposure, yet many of these donors possessed B_mem_ cells in high frequencies. Tangentially and in agreement with recent reports supporting the reactivation of EBV in COVID-19 patients [75,76], our antibody binding data revealed higher levels of EBNA1-reactive IgG in the COVID-19 samples relative to the pre-COVID-19 samples, which was maintained even when the cohorts were subcategorized according to sex, age, or race/ethnicity (Appendix A). Collectively, the findings made for EBNA1 further support the notion that monitoring the B cell memory compartment itself can be a more sensitive and reliable indicator of immune exposure and memory potential than standard measurements of antibody reactivities alone.

**Figure 4 cells-11-03662-f004:**
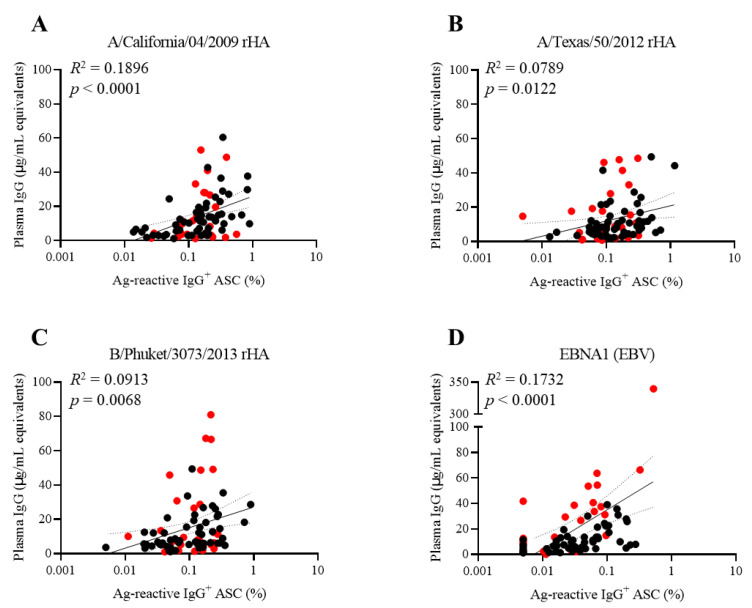
Discordance between circulating antibody levels and memory B cell frequencies in PBMCs for seasonal influenza viruses and EBV. Subjects from the pre-COVID-19 (*n* = 54, black dots) and COVID-19 (*n* = 25, red dots) donor cohorts were evaluated for IgG^+^ ASCs and circulating IgG reactivity against (**A**) recombinant hemagglutinin (rHA) representing the seasonal H1N1 influenza virus A/California/04/2009, (**B**) rHA representing the seasonal H3N2 influenza virus A/Texas/50/2012, (**C**) rHA representing the seasonal influenza B virus B/Phuket/3073/2013, or (**D**) EBV EBNA1. The frequency of antigen-reactive IgG^+^ ASCs is expressed as a percentage of the total IgG^+^ ASC compartment (x-axis), and the levels of circulating antigen-reactive IgGs are plotted as “μg/mL equivalents” (y-axis) for each subject.

### 3.6. High Reproducibility of Total and Antigen-Reactive IgG^+^ ASC Frequency Measurements

While we had limited numbers of PBMCs available for most subjects in the COVID-19 cohort, cells obtained from pre-COVID-19-era donors were available in sufficient quantities to assess the reproducibility of either total or antigen-reactive IgG^+^ ASC frequency measurements. We tested replicate aliquots of PBMCs cryopreserved from the same blood drawn in two separate experiments. Thus, except for the cryopreserved cell material being the same, all other steps of the testing process were independent assay variables, including (a) thawing and washing of the cryopreserved cells, (b) live/dead cell counting, and adjusting the PBMC concentration to 2 × 10^6^ cells/mL for the (c) subsequent polyclonal stimulation during the five day cell culture period, followed by renewed live/dead counting of the PBMCs and readjusting their concentration for being plated into (d) the FluoroSpot assays (total or antigen-specific). The (e) visualization and (f) counting of the SFUs also represent potential inter-assay variables. In this way, for each pre-COVID-19-era donor, frequencies for (g) the total IgG^+^ ASCs as well as (h) the frequency of the antigen-reactive IgG^+^ ASCs were determined, allowing for the calculation of “% antigen-reactive IgG^+^ ASCs relative to the total IgG^+^ ASCs”. Appendix A summarizes the results obtained from the repeat testing of the total IgG^+^ ASC frequencies using replicate vials of PBMCs, while Appendix A depict the reproducibility of antigen-reactive IgG^+^ ASC frequencies established against the CA/09 rHA, TX/12 rHA, and EBNA1 proteins. Collectively, these data show that despite the multitude of potential assay variables, the frequencies of ASCs reactive with these antigens reproduced closely.

### 3.7. Discussion of the Mechanism Underlying Discordance and Implications

Our current understanding of the fate decision checkpoints determining whether an activated B cell will join the B_mem_ cell compartment or differentiate into an antibody-secreting PC is the subject of several excellent reviews [1,16,77,78,79,80]. While many parameters can contribute to B cell fate decisions after antigen encounters, in the following, we will focus on just two mechanisms that could account for or contribute to the discordance between circulating antibody levels and antigen-reactive B_mem_ cell frequencies.

First, limitations on PC versus B_mem_ cell survival might account for our findings. PCs can be classified into two types: those that are short-lived and contribute to antibody titers only acutely and those that are long-lived (LLPC) and provide sustained antibody production for decades and potentially the lifetime of an individual [81,82,83]. Presently, there is an incomplete understanding of what distinguishes the ability of a LLPC to survive relative to a short-lived ASC [2]. Importantly, LLPCs are not intrinsically long-lived, and instead, their survival is dependent on the acquisition of a distinct transcriptional profile and their ability to access specialized and pro-survival niches such as those existing in the BM. Furthermore, there is likely only a finite number of suitable niches in such anatomical locations in which PCs can take up residence and acquire longevity through their intimate interactions with stromal cells and receipt of pro-survival cues [2,84,85]. As such, a plausible explanation for the reduced levels of antigen-reactive circulating antibodies detected in many of the subjects investigated in this communication is that GC-derived ASCs were in fact generated as a consequence of the virus infection but failed to successfully take up long-term residence in the BM. B_mem_ cells, in contrast, do not need to compete for such niches for their survival. While in theory the ImmunoSpot^®^ assay would be particularly well-suited to directly test this hypothesis by enumerating individual subjects’ PC frequencies in the BM [86] versus B_mem_ cell numbers in the blood, the BM compartment is not readily accessible for routine immune monitoring, and thus assessment of LLPCs in our donor cohort was not possible. While this mechanism might explain why antibody levels can be low in the presence of abundant B_mem_ cells, it does not account for the reverse scenario.

The differential, affinity-based selection of mutated B cell subclones along the PC cell versus the B_mem_ cell differentiation pathways might account for our findings. The GC is an anatomical site in which antigen-activated B cells can undergo multiple rounds of cell division and progressively acquire somatic hypermutations in their BCR’s *IgH*/*IgL*. Perhaps most relevant to the interpretation of the data presented here is the well-established notion that the fate decision of a given GC B cell subclone is determined based on the affinity of its BCR for the eliciting antigen [1,16]. Namely, GC B cell subclones endowed with a low-affinity BCR for the antigen stop proliferating (especially at later stages of the response when the antigen becomes limiting) due to insufficient interactions with follicular T helper (T_FH_) cells, and instead exit the GC reaction and become long-lived B_mem_ cells. In contrast, GC subclones possessing a high-affinity BCR are instructed by T_FH_ cells to undergo further rounds of proliferation and acquire additional somatic hypermutations [87,88]. As antigens become increasingly limiting in the GC reaction, the pressure for selection of high-affinity GC B cell subclones becomes even more stringent, while the remainder of the B cells with a lower affinity for the antigen are shunted towards the B_mem_ cell pool. This affinity-based selection process, often referred to as affinity maturation, ultimately leads to the differentiation of PCs producing antibodies with an exquisitely high affinity for the eliciting antigen and B_mem_ cells of high, but primarily lower affinities [89]. Consequently, GC-derived PC and B_mem_ cells represent two related, yet nonoverlapping, antigen-reactive repertoires with divergent cell fate decisions based on their BCR’s affinity for the antigen.

Therefore, it is assumed that B cell responses that give rise to GC reactions build “two walls of protection against pathogens” [1]. The first wall, encompassed by pre-formed antibodies possessing high affinity, affords the advantage of providing instant defense against the homologous pathogen, but on the downside, these antibodies can potentially interfere with the ability to respond to new variants that are antigenically similar [90,91]. The second wall of antibody-mediated defense is comprised by B_mem_ cells that not only can participate in anamnestic responses against the homologous virus, but also contribute clonally expanded, class-switched, and semi-affinity-matured precursor cells that may be reactive with emerging variants of the pathogen, which can undergo further affinity maturation to fine tailor the B cell response to such variants [92,93].

The above notion of divergent affinity-based selection underlying alternative PC cell versus B_mem_ cell repertoires has primarily arisen from studies of murine models involving, by necessity, minimalist approaches such as the use of BCR-transgenic mice and model antigens [94]. To what extent they are applicable to human B cell responses against viruses in large remains unverified. These murine studies were further enabled by the reliance on variable-reducing test conditions, including the homogenous genetic background, age, and sex of the mice studied, their controlled specific-pathogen free environment, defined antigen exposure, and through unfettered access to the lymphoid tissues in which B cell immune responses evolve. As indicated previously, human PCs residing in the BM are not readily accessible for general immune monitoring purposes. While human B_mem_ cells can be readily accessed via blood drawing, the ability to study them systematically in larger cohorts awaits an enabling technology. To our knowledge the data communicated here represent the first systematic study comparing virus-reactive B_mem_ cell frequencies with circulating antibody levels in sizable human cohorts. The basic finding that antibody levels and B_mem_ cell frequencies are frequently discordant may not come as a surprise in face of the murine literature; however, the extent of the inter-individual variations between serum antibody and B_mem_ cell frequencies we observed in our human cohorts was, to a degree, unexpected. All of our PBMC donors were healthy young or middle-aged adults, and all the B cell responses we studied were induced by and directed against viruses that these adults successfully controlled. However, in some of these individuals, memory B cell frequencies were remarkably high, while antibody titers were surprisingly low. In the context of our own dataset, host factors such as sex, age, race/ethnicity, or length of time between the SARS-CoV-2 infection and the sample acquisition could not account for the discordance between circulating antibody levels and antigen-reactive B_mem_ cell frequencies (summarized in Appendix A). Therefore, we postulate that there might be an evolutionary pressure behind developing such discordant phenotypes. The affinity-based differential selection of B_mem_ cells versus PCs [1] along with stochastic fate decisions for rare antigen-specific precursor cells [95,96] may explain the discordant engagement of the first versus the second wall of humoral immune defense in individuals

With fundamental pros and cons for pre-existing, antibody-afforded versus B_mem_ cell-mediated protection, there could exist an evolutionary advantage for endowing different individuals in the population with the propensity to preferentially rely on one or the other immune defense strategy. While with the viruses we studied here, either strategy is compatible with a successful defense, it can be envisioned that with some pathogens, the fate decisions of B cells could have profound implications on the outcome of the host.

### 3.8. Concluding Remarks

Our data also have fundamental implications for immune monitoring in general. Measuring serum/plasma antibodies only, as it is commonly conducted in clinical trials and in the clinic, provides information only on the abundance and efficacy of the “first wall” of antibody-mediated defense. A better understanding of the “second wall” of adaptive immune defense, encompassing the B_mem_ potential against both homo- and heterotypic viruses, in contrast, requires direct assessment of the B_mem_ cells themselves. In support of this notion, we found that elevated frequencies of antigen-reactive IgG^+^ B_mem_ cells are a more sensitive measure for revealing past SARS-CoV-2 virus exposures than antibody positivity (see Figure 2C and Figure 3C and the manuscript in preparation). Towards the practical end, we showed the feasibility of the high-throughput assessment of antigen-reactive B_mem_ cells, enabling their inclusion in the routine immune monitoring portfolio.

## Figures and Tables

**Figure 1 cells-11-03662-f001:**
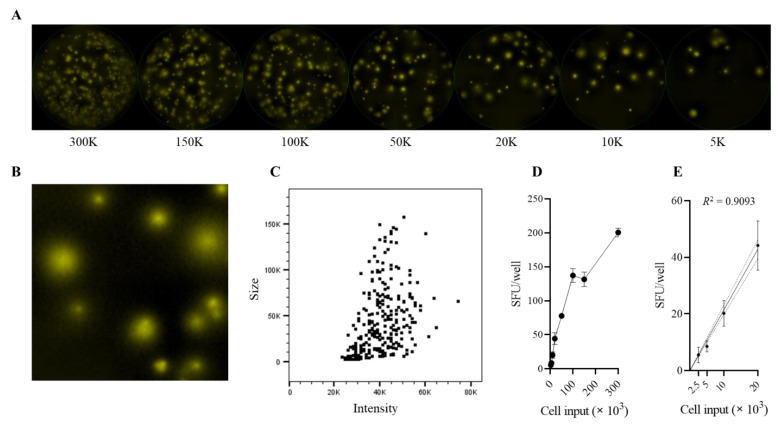
Accurate measurement of SARS-CoV-2 spike (S1)-reactive ASC frequencies. PBMCs from COVID-19 Donor 1 were pre-stimulated for 5 days in vitro to transition resting B cells, irrespective of their specificity, to antibody-secreting cells (ASCs). PBMCs were then seeded into SARS-CoV-2 S1-coated wells with decreasing cell inputs, starting at 3 × 10^5^ PBMC/well, with increasing numbers of replicate wells. During the overnight culture period, antibodies originating from S1-reactive ASCs were captured on the antigen-coated membrane in close proximity, and the resulting antibody secretory footprints, or spot-forming units (SFUs), were then detected as described in Materials and Methods. (**A**) Representative well images are shown for the specified cell numbers plated per well (an overview image depicting all replicate wells for COVID-19 Donor 1 is shown in Appendix A). (**B**) Magnification of a representative well image seeded with 2 × 10^4^ PBMC/well in which individual SFUs are clearly discernable and exhibit variable sizes and fluorescent intensities. (**C**) SARS-CoV-2 S1-reactive secretory footprints, originating from replicate wells seeded with 2 × 10^4^ PBMCs, were merged into a flow cytometry standard (FCS) file and visualized as a bivariate plot depicting the fluorescence intensity (x-axis) and size (y-axis) of the individual FluoroSpots. (**D**) Mean ± SD of SARS-CoV-2 S1-reactive SFU counts as a function of cell input. Note the deviation in linearity for SFU counts originating from wells seeded with greater than 2 × 10^4^ PBMC/well inputs. (**E**) Linearity of SARS-CoV-2 S1-reactive SFU counts originating from wells seeded with less than 2 × 10^4^ PBMC/well. Extrapolation of the regression curve was used to establish the frequency of S1-reactive IgG^+^ B cells in COVID-19 Donor 1 at 195 SFU/10^5^ PBMC.

**Figure 2 cells-11-03662-f002:**
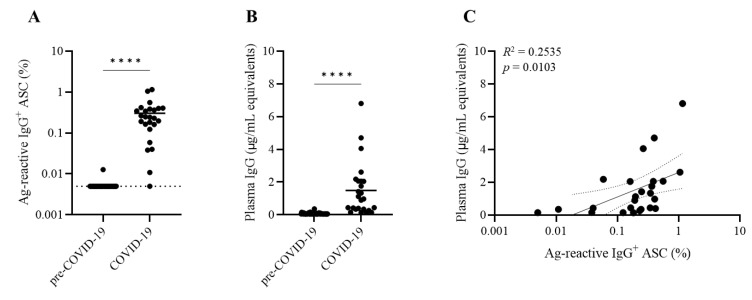
Specificity of SARS-CoV-2 spike (S1)-reactive IgG^+^ memory B cell detection and discordance between circulating antibody levels and memory B cell frequencies in PBMCs. (**A**) SARS-CoV-2 S1-reactive IgG^+^ ASC frequencies in PBMCs from pre-COVID-19 (*n* = 54) or convalescent COVID-19 donors (*n* = 25). Donor PBMCs were pre-stimulated for 5 days in vitro and were seeded into SARS-CoV-2 S1-coated wells as described in Appendix A and Materials and Methods. For each donor, represented by an individual dot, the frequency of the S1-reactive ASCs is expressed as the percentage of the total IgG^+^ ASC compartment in that subject. Dotted line denotes limit lower limit of quantification. Statistical significance was determined using an unpaired Student’s *t*-test. **** *p* < 0.0001. (**B**) SARS-CoV-2 S1-reactive circulating IgG antibody levels in pre-COVID-19 and COVID-19 donors were measured by ELISA (detailed in Materials and Methods) alongside an IgG reference standard that allowed for the interpolation of S1-reactive IgG levels as “μg/mL equivalents”. Donors are represented by an individual dot. Solid line denotes mean IgG level of the COVID-19 donor cohort. Statistical significance was determined using an unpaired Student’s *t*-test. **** *p* < 0.0001. (**C**) Correlation between SARS-CoV-2 S1-reactive IgG^+^ ASC frequencies (x-axis) and circulating IgG levels (y-axis) in donors comprising the COVID-19 cohort.

## Data Availability

The data generated in this study will be made available by the authors, without undue reservation, to any qualified researcher.

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
