# Peer review of "Antibody Levels Poorly Reflect on the Frequency of Memory B Cells Generated following SARS-CoV-2, Seasonal Influenza, or EBV Infection"

_cells, 2022, doi:10.3390/cells11223662_

Round 1
Reviewer 1 Report
Remarks to the Author:
Carla Wolf and colleagues used an improved antigen-specific B cell ImmunoSpot® assays, which has high affinity of membrane coated recombinant protein with 6-His tag, to detect Bmem cell frequencies for viral infections such as SARS-CoV2, influenza and EBV. They found that the relationship between antibody levels and Bmem cell frequencies differ in terms of the duration and periodicity of antigen exposure. It is important that they show the B cell memory function is irrelevant with pre-formed humoral immunity in anti-viral immunity. The method is much useful for detecting different antigen-reaction Bmem cells. The writing is very logical. The data presentation is very reasonable
Major points:
The SARS-CoV-2 S1-reactive Bmem cell seems as 0.1-0.2% of PBMC. Is it as high as physiology condition or is it because PBMC is stimulated by polyclonal?
Why not use sorted B cells?
How about the frequency of SARS-CoV-2 S2-reactive Bmem cell as S2 also can induce antibody production by B cell?
Traditional Bmem cells expresses high level IgM and IgD, they just use a human IgG-detecting kit. How about the frequency of these memory B cells?
Minor points:
It is better to short the instruction of B cell ImmunoSpot assay.
The writing is too long and needs to be cut down.
Author Response
We thank Reviewer 1 for his/her overall positive assessment of our manuscript, including the judging: “The method is much useful for detecting different antigen-reaction Bmem cells. The writing is very logical. The data presentation is very reasonable”.
The following Major points have been raised:
Critique 1. “The SARS-CoV-2 S1-reactive Bmem cell seems as 0.1-0.2% of PBMC. Is it as high as physiology condition or is it because PBMC is stimulated by polyclonal?”
Answer to Critique 1: The Y axes of these figures have been labelled “Ag-reactive IgG+ ASC (%)”: the Reviewer is correct; however, we did not spell this out clearly enough in the text. In response to this valid point, we now specify in the Materials and Methods section of the revised manuscript: “Following convention in flow cytometry, the data are expressed as the percentage of antigen-reactive SFU among all IgG+ ASC” (Lines 269-270). Of note, as IgG+ ASC constitute between 1 and 20% of viable PBMC following the polyclonal stimulation protocol (please see Figure S7B), a frequency of 0.1% among IgG+ ASC corresponds to 0.02-0.001% among PBMC. Also, please also note that the frequencies of S1 antigen-reactive B cells that we have detected by ImmunoSpot are consistent with reports in which such assessments were done by flow cytometry (PMID 34610291 and PMID 34843448).
Critique 2: Why not use sorted B cells?
Answer to Critique 2: The Reviewer is correct; we should have addressed this issue. We did it in the revised manuscript, adding the following statement (Lines 117-120). “Another advantage of ImmunoSpot assays is that they can be done with unseparated PBMC, without the need for prior B cell isolation/sorting as B cells are the sole source of secreted antibody in PBMC (the need for B cell enrichment/sorting would significantly constrain assay throughput for immune monitoring purposes).”
Critique 3: “How about the frequency of SARS-CoV-2 S2-reactive Bmem cell as S2 also can induce antibody production by B cell?”
Answer to Critique 3. This is an excellent question. To address it, we added to Materials and Methods the following statement (Lines 238-240): “Our objective was to monitor antibody and B cell responses that would be specific to SARS-CoV-2. Since the S2 subunit (S2) is more conserved between seasonal coronaviruses and SARS-CoV-2, we focused on the S1 domain.” Of note, we have since evaluated several donors from both the pre-COVID era and PCR-verified donor cohorts for antibody and memory B cell reactivity against a full-length Spike antigen (Spike FL) and these data will be included in a future manuscript. Relevant to this Reviewer’s comment, little if any antibody or memory B cell reactivity against the Spike (FL) antigen was detected in the pre-COVID era donors which indicates that S2-reactive antibody levels and memory B cell frequencies are at or below the limit of detection of these assays.
Critique 4: “Traditional Bmem cells expresses high level IgM and IgD, they just use a human IgG-detecting kit. How about the frequency of these memory B cells?”
Answer to Critique 4. While the Reviewer is correct stating that IgM (but not IgD) molecules can be expressed by a subset of memory B cells, these memory B cells have, for the most part, not undergone somatic hypermutations and affinity maturation. We attempted to detect such S1 antigen-reactive IgM secreting memory cells in our own studies following polyclonal stimulation of donor PBMC but could not readily distinguish S1-reactive IgM+ ASC from donor intrinsic poly-reactivity (we could provide the data to the Reviewer if he/she wishes). But again, the Reviewer is correct, we should have clarified why we focused on IgG, which we now did adding to Materials and Methods (Lines 255-256): “Unlike memory B cells that express an IgM BCR, the memory B cell population that primarily participates in recall responses has already undergone class-switch recombination and is programed to secrete IgG (Ref 1). In this study, we therefore focused on the IgG expressing memory B cell population.”
Critique, minor points: “It is better to short the instruction of B cell ImmunoSpot assay. The writing is too long and needs to be cut down.”
Answer: To put our data in perspective, we had to bridge the highly specialized field of basic B cell immunology with that of the sero-diagnostic oriented field of clinical immune monitoring. We are glad to have succeeded with this task as this Reviewer (concurring with Reviewer 2) judged overall: “The writing is very logical. The data presentation is very reasonable”. In response, we have cut the following lines from the original manuscript: 42-47, 60-66, and 7-90.
Reviewer 2 Report
In this paper, the authors established and improved a new assay to quantify peripheral blood memory B cells specific for SARS-CoV2, seasonal influenza and EBV, and found no correlation between corresponding plasma antibody titers and B mem cells against the same virus. The paper is well written, the methods are well described, the results are very interesting and surprising and the conclusions are important
The introduction is very exhaustive but could be a little bit shorter.
Author Response
We thank Reviewer 2 for his/her overall positive judgement of our manuscript, concluding: “The paper is well written, the methods are well described, the results are very interesting and surprising and the conclusions are important”.
Reviewer 2’s only point of critique is: “The introduction is very exhaustive but could be a little bit shorter”, In response, we have cut the following lines from the original manuscript: 42-47, 60-66,7-90.
Reviewer 3 Report
Authors of this manuscript studied and compared plasma antibodies with memory B cells generated after SARS-CoV-2, seasonal influenza or EBV infections. They found that antibody levels induced after these infections are poorly correlated with the frequency of memory B cells. The findings are important for many clinical and vaccine studies. The study design is proper, the study size is reasonable. Overall the manuscript is well written. I have only a few suggestions.
1. The introduction is pretty long, and could be shortened.
2. Lines 197-205, it would be important to mention the time when the convalescent COVID-19 donors were recruited, although the detailed info was included in the supplementary data. This is because variants of SARA-CoV-2 emerged fast. Authors should discuss how their finding is relevant for different variants.
3. Line space should be uniform through the manuscript.
Author Response
We thank Reviewer 3 for his/her overall affirmative judgement, concluding: “The findings are important for many clinical and vaccine studies. The study design is proper, the study size is reasonable. Overall the manuscript is well written. I have only a few suggestions.”
Critique 1: The introduction is pretty long, and could be shortened.
Thanking the Reviewer for this suggestion, we have substantially shortened the introduction.
Critique 2: Lines 197-205, it would be important to mention the time when the convalescent COVID-19 donors were recruited, although the detailed info was included in the supplementary data. This is because variants of SARA-CoV-2 emerged fast.
We thank the Reviewer for this suggestion. In the revised manuscript (Lines 201-207) we now specify that “The SARS-CoV-2 infection-verified blood samples were collected between April and October 2020. However, based on PCR testing these infections occurred in the months of April, May and June when the initial Wuhan strain virus was circulating and prior to the emergence of new SARS-CoV-2 variants or availability of COVID-19 vaccines. Accordingly, we evaluated B cell and circulating antibody reactivity against the Spike protein encoded by the Wuhan strain virus. This simple, defined, immunologic scenario was chosen to avoid the introduction of further immune variables caused by repeat infections with partially cross-reactive heterotypic virus variants in a mostly (homotypic Spike antigen) vaccinated population in which re-infections frequently are subclinical or undiagnosed.”
Critique 3. Authors should discuss how their finding is relevant for different variants.
We agree with the Reviewer that it will be important to establish how homotypic vs. heterotypic antigen-reactive memory B cell frequencies vs. circulating antibody levels are shaped by infections/reinfections with new variants in a mostly homotype-vaccinated population, but even discussing these complexities in due depth would exceed the scope – and space limitations- of this manuscript, also considering all Reviewer’s comments that the manuscript already is too long.
- Line space should be uniform through the manuscript.
We thank the Reviewer for bringing this oversight to our attention. We have made spacing uniform in the revised manuscript.
Round 2
Reviewer 1 Report
-
They answered my questions and made changes very well.